# The influence of the U.S. export controls against China on the resilience of Chinese corporates

**Huan He[1]◉, Peng Chen[2]◉, Xianjing Huang[3]*, Le Li[4]◉**

1 School of Finance and Insurance, Guangxi University of Finance and Economics, Nanning, Guangxi, China, 2 CHINA-ASEAN School of Economics, Guangxi University, Nanning, Guangxi, China, 3 School of Finance And Insurance, Guangxi University of Finance and Economics, Nanning, Guangxi, China, 4 School of Finance And Insurance, Guangxi University of Finance and Economics, Nanning, Guangxi, China

◉ These authors contributed equally to this work.
* xianjing_h@163.com

## Abstract

As Sino-US trade frictions intensify, the United States has continually strengthened its export control measures against China, posing potential risks to the international supply chains and production operations of Chinese enterprises. This paper has compiled data from the U.S. Bureau of Industry and Security (BIS) Entity List and employed A-share manufacturing listed companies in China from 2015 to 2023 as research samples. By treating the U.S. export controls against China as a quasi-natural experiment, we employ a multi-period difference-in-differences model to analyze the impact of these export controls on the resilience of Chinese manufacturing listed companies and the underlying mechanisms. Our findings reveal that U.S. export controls significantly reduce the resilience of affected firms within the industry in China, primarily by inhibiting internationalization processes, exacerbating financing environments, and intensifying corporate risks. Further analysis shows that firms with higher product market competitive advantages, greater analyst coverage, and more aggressive development strategies are better equipped to mitigate the shocks of export controls. Conversely, firms lacking political connections or operating in environments with higher external uncertainties are more susceptible to negative impacts, with their resilience significantly diminished. This study broadens the research scope of international trade policies and corporate resilience, offering insights for Chinese enterprises to enhance their adaptive capabilities and optimize business strategies in the complex and volatile international trade environment.

## 1. Introduction

In the wave of globalization and digitization, the complexity of international supply chains and their interdependence have reached an unprecedented level. This trend has substantially influenced the advancement of the global economy while

**Data availability statement:** Corporate resilience serves as the key outcome variable in our analysis. The growth of enterprise's cumulative sales revenue in three years and standard deviation of stock's monthly return are used to build corporate resilience indicator. Raw data comes from third-party database called China Securities Market and Accounting Research Database (CSMAR database, https://data.csmar.com/ e-mail address: service@csmar.com). Data on U.S. export controls are sourced from the official website of the U.S. Bureau of Industry and Security (BIS). Chinese listed companies included in the 2018–2023 entity lists are manually screened. Firms are classified using the standardized industry taxonomy outlined in the 2012 Listed Company Industry Classification Guidelines issued by China's CSRC. The original data of key explanatory and control variables are obtained from the China Securities Market and Accounting Research Database and Wind Database. Authors further measured them. The explanatory variable corporate resilience, Firm size (Size), Leverage (Lev), Age at listing (Inage), Return on Equity (ROE), Growth, Current Ratio (Liqtratio), Cash holdings (Cash), Price-earnings ratio (PE), Proportion of independent directors (Independ), Chairman's ownership of shares (Chair), Nature of equity (SOE), Herfindahl Index (HHI), Firms' innovation capacity, Firms' government grants, total financing inflows (debt + equity + related-party financing) scaled by total assets (Finasset), the ratio of bank borrowings (long- and short-term) to total assets (banklev), Firm stock crash risk , Firm downside risk, Firm stock crash risk variables, Firm downside risk variables, Product market competitive advantage, Analyst attention, Political affiliation, Corporate strategic aggressiveness, Customer concentration and Supplier concentration are from the China Securities Market and Accounting Research Database (CSMAR database, https://data.csmar.com/ e-mail address: service@csmar.com). The original data of breadth of corporate internationalisation structure (Overfirm) and intensity of firms' internationalization (Oversale) are from the WIND database (https://www.wind.com.cn/portal/zh/EDB/index.html e-mail address: GELSupport@wind.com.cn). The original data on environmental uncertainty are from the China Economic Policy Uncertainty Index and are further measured by authors (https://fred.stlouisfed.org/series/CHNMAINLANDEPU link: https://fred.stlouisfed.org/contactus/) All other

simultaneously creating various emerging challenges and risks. With the changes in the geopolitical landscape and the intensification of technological competition, U.S. policies for export control have continued to evolve, with extensive effects on the global supply chain and international trade landscape. In 2018, the U.S. passed the Export Control Reform Act (ECRA), which adjusted its foreign export control policies for key technology areas such as semiconductor chips, artificial intelligence, and quantum computing. Technological competition drove a strategic shift in U.S. export control policy with the introduction of the Act. With China's development in AI and semiconductor manufacturing, the U.S. has included AI chips and advanced process equipment (e.g., EUV lithography) in its export restrictions. As technological competition has grown, the United States' export controls on China have become much stricter since 2018. By the end of 2024, more than 1,800 Chinese companies have been sanctioned by the United States (S1 Fig).

This series of regulatory measures has posed unprecedented challenges to enterprises, including narrowing procurement channels for key raw materials, impeding overseas market expansion, surging compliance costs, and the heavy pressure of technological embargoes. These negative effects have significant contagion effects, directly affecting the sanctioned firms and those within the identical supply chain [1] and the entire industry ecosystem. In particular, the firms highly dependent on the unimpeded flow of global supply chains and access to international markets, whose survival and development are being severely tested. Against this background, enterprise resilience has become an important factor in determining survival. Firms with strong resilience can quickly adjust their strategies, find alternative resources and markets, and innovate their technologies and management models to achieve sustainable development under the pressure of export controls [2]. Therefore, an in-depth study of the consequences of U.S. foreign export regulations on enterprise resilience is of vital practical significance. It enhances comprehension of the growth of the international political and economic landscape while offering crucial theoretical and useful direction for firms to achieve sustainable development in a complex and dynamic setting.

Scholars have mainly elaborated on the impact effects of export controls from the macro and micro perspectives. Export controls, as delineated by the Export Administration Regulations (EAR), are mechanisms the U.S. government employs to advance national security interests and foreign policy goals by regulating sensitive equipment, software, and technology exports. At the macro level, scholars mainly focus on exploring the influence of export controls regarding the economic performance of the target nation, the international economic and financial markets, and the currency exchange rate. Research indicates that the enforcement of economic sanctions can adversely affect the economic relations between the sanctioning nation and the targeted nation [3,4] and inhibit bilateral trade [5]. In particular, economic sanctions have significantly dampened economic growth in target countries [6,7,8]. This effect is even more pronounced in developing and emerging market countries [9], while government mismanagement of a country's economy will make it highly vulnerable to economic sanctions. These sanctions can be implemented by directly attacking key

data necessary to replicate the results of the study in their entirety are contained within the paper and its Supporting Information files.

**Funding:** Funding provided by the National Social Science Fund of China (22XJY002) to XH, Guangxi Philosophy and Social Science Planning Research Project (22CJY006) to XH, the Young Talent Funding Program of Guangxi to HH, the Project for Enhancing Basic Research Capabilities of Mid-aged and Young Teachers in Guangxi Universities (2025KY0628) to HH, and Guangxi First-class Discipline Applied Economics Construction Project Fund to HH.

**Competing interests:** The authors have declared that no competing interests exist.

sectors such as government leadership, state-owned banks, and energy companies [10]. At the same time, these sanctions can also be implemented through indirect means, such as cutting off overseas financial flows, reducing foreign direct investment, and worsening the financing conditions of the target country's economy [11]. In addition, U.S. export sanctions can have significant spillover effects on the global stock market and foreign exchange market, exacerbating exchange rate volatility in target countries [12], reshaping the pattern of global supply chains, and triggering a "redistribution of economic interests" among countries [13].

At the micro level, scholars have mostly focused on studying the impact of export control on the innovation, investment, and operational efficacy of high-tech firms in designated countries. The technology blockade imposed by the United States on China through restrictions on inward investment and export controls will affect normal economic and trade exchanges and the cost of technology learning for Chinese enterprises [14]. Consequently, export limits will adversely affect the innovation of Chinese enterprises in the short term, mostly by diminishing the quality of imported items, constricting the extent of international operations, and curtailing R&D spending. Nonetheless, in the long term, it will exert a positive influence on the autonomous innovation of Chinese firms [15–17]. Shou and Tan [18] found that entity list policies significantly dampen firms' investment levels, with a more pronounced effect on non-state-owned enterprises, firms with high fixed asset ratios, and highly capital-intensive firms. Initially, entity list occurrences adversely affect business performance significantly. In contrast, the degree of negative effects diminishes over time, and even positive feedback has been observed in some events in recent years [19].

In summary, most current research examines the multifaceted effects of U.S. export sanctions on target countries' macroeconomic and international financial markets, as well as preliminary micro studies of enterprise innovation, investment, business performance, etc. However, research on the mechanisms and variety of the effects of export controls at the enterprise level, particularly regarding company resilience, remains insufficient. Existing research has defined business resilience as an important ability of a business entity to cope with external shocks and secure long-term growth [20]. From the viewpoint of the current state of empirical research, although there has been a gradual increase in research on trade policy and enterprise resilience in recent years. However, there is still a significant gap in the systematic empirical analysis of the mechanism of the effect of export controls on firm resilience and the issue of heterogeneity. Existing literature mostly focuses on the direct impact of a single dimension but lacks a systematic disentanglement of the complex transmission path between export controls and firm resilience. While a few studies have examined variability in business ownership, size, or area, their conclusions predominantly rely on static cross-sectional data and do not capture variations in firms' dynamic responses across the policy cycle. As an important indicator of the ability of enterprises to cope with external shocks in a turbulent environment, enterprise resilience has a crucial effect on the sustained advancement of enterprises. Enterprises with high resilience can respond positively to changes in the external environment, turning risks into opportunities and realizing counter-trend growth.

Enterprises with low resilience are often reshuffled in the face of external environmental shocks due to a shortage of resources and are caught in a dilemma [21,22]. Consequently, amid the prevailing economic uncertainty, it is crucial and imperative to thoroughly explore the effects of U.S. export sanctions on corporate resilience and the mechanisms that underlie these effects.

This paper aims to rigorously examine the effects and internal mechanisms of U.S. foreign export control on the resilience of Chinese firms, utilizing a sample of Chinese A-share manufacturing companies listed on the Shanghai and Shenzhen Stock Exchanges. This paper further breaks down the research perspective. It explores how variables such as market competitive advantage, analyst attention, political affiliation, and strategic aggressiveness of firms act as moderators on how they affect the relationship between export controls and firm resilience. Heterogeneous responses exhibited by different firm traits responding to trade sanctions are revealed. Compared to prior studies, this paper's potential innovations and contributions mostly lie in the following areas. First, it broadens the study of the micro effects of export controls. Most of the current research on the impact of U.S. export sanctions focuses on theoretical studies or studies on macro effects. At the same time, there are relatively few studies at the firm level. This work expands the research scope in this domain and offers a novel perspective on the influence of U.S. export limits on company behavior. Second, it enriches the theoretical and practical framework of firm resilience research. The existing literature has studied the factors influencing corporate resilience regarding the macro and micro environments firms face, firm characteristics, and managerial characteristics. However, there is still insufficient research on how external extreme events such as export sanctions play a specific role in corporate resilience. This paper fills this research gap by including export sanctions in studying corporate resilience. At the same time, it also offers significant theoretical assistance and practical direction for firms to bolster their resilience and adeptly manage external shocks amid the turmoil of globalization. Thirdly, it clarifies the mechanism between export control and enterprise resilience and enriches the empirical research on the transmission path in the existing literature. The paper clarifies the mechanism of influence of U.S. export restriction policy on enterprise resilience and provides practical strategies and directions for enterprises to deal with U.S. export control.

## 2. Analysis of theoretical mechanisms and research hypotheses

Despite the gradual increase in research on trade policy and enterprise resilience in recent years, there is still a paucity of research on how export controls affect enterprise resilience. Researchers examining the effects of trade policy uncertainty concentrate on the implications of tariffs, export subsidies, export controls, import controls, and other policies on firms. Trade policy uncertainty significantly increases firms' financialization [23], reduces firms' investment in innovation and financial investment [24,25], and significantly negatively affects trade credit and firm value [26]. It also substantially inhibits the breadth and depth of firms' exports [27] and impairs firms' operational performance [28]. Focusing on export controls, scholars have studied the effects of export restrictions on enterprises, mainly in terms of corporate ESG and corporate innovation. Liu et al. examined the impact of the U.S. export control policy on firms in sanctioned industries that are not blocklisted, revealing that blocked enterprises within an industry markedly enhance the ESG performance of other firms in that sector [29]. Yu et al. found through empirical tests that export control hurts corporate innovation in the short term and can have a forcing effect on corporate independent innovation in the long term [30]. In the research related to corporate resilience, scholars have examined the determinants of corporate resilience mainly from both macroeconomic and microeconomic viewpoints. From a macro point of view, the enhancement of governmental assistance policies [31] and the degree of investor protection [32] both enhance enterprise resilience. At the micro level, firms' level of digitization [33], innovation capability [34], and analytical capability [35] are positively associated with firm resilience. Most scholars explore trade policy's influence on enterprises but neglect resilience, a crucial characteristic for firms. Can export controls have an impact on firm resilience? If so, through what channels do export controls affect firm resilience? This part provides an in-depth study of this issue and proposes relevant theoretical hypotheses to explore further the impact mechanisms between export controls and firm resilience.

## 2.1 Export controls and business resilience

According to the theory of comparative advantage, countries should focus on producing products with a comparative advantage and achieve optimal allocation of resources and maximization of welfare through international trade. However, export controls may upset this balance of comparative advantage, causing otherwise competitive countries to lose their advantage in certain areas, thereby affecting their overall economic performance and potential for industrial upgrading. China has a significant comparative advantage in many manufacturing sectors due to its low labor costs and well-developed industrial support system. Simultaneously, the United States is unique worldwide due to its deep accumulation of high-tech fields and technological innovation capabilities. However, the U.S. export control has broken this normal trade pattern. Take the electronic information industry as an example. Chinese enterprises have significant advantages in the assembly and manufacturing process. They depend highly on the United States for high-end chips and other core technology products. Implementing export controls has cut off this key link in the supply chain, forcing Chinese companies to face a shortage of key products. Production plans originally based on comparative advantages have been forced to adjust, and companies have had to switch to more expensive alternatives with poor performance stability. This has directly led to a decline in production efficiency and a sharp increase in the threshold and cost of technology introduction [36]. Fiberhome Communications, a leader in China's communications industry, was placed on the Entity List by the U.S. Department of Commerce on May 22, 2020 (S3 File). Export restrictions have constrained Fiberhome's access to essential components and technologies from the United States. Enterprises face component shortages, supply chain disruptions, and production process limitations regarding production efficiency. Meanwhile, it forces enterprises to increase the difficulty of technology acquisition, the cost of alternative technologies, and the pressure of long-term R&D in terms of technology cost [37]. These effects challenge firms' short-term operations and create many uncertainties about their long-term development and competitiveness enhancement. Ultimately, Fiberhome's stock price plummeted, operating income and profit declined, and financial risks rose sharply (Specific cases are listed in Supporting information 3).

Firm resilience measures firms' ability to cope with changes in internal or external environments, integrate resources, recover quickly, and sustain growth [38]. It has been severely tested under the impact of export control. Restrictions on imports of key components and technologies not only limit the expansion of enterprise production scale but also may trigger the shrinkage of export markets, the sharp loss of economies of scale, unit costs, and profit margins. In the face of survival pressure, enterprises are often forced to take short-term response measures, such as cutting R&D investment, downsizing staff, etc. Although these initiatives can temporarily alleviate the urgent need but fundamentally weaken the enterprise's innovation ability, market competitiveness, and long-term development potential, enterprise resilience is seriously weakened. Nonetheless, enterprises find mitigating the enduring adverse impacts of export prohibitions challenging if they depend solely on immediate responses. To remain competitive in a complex international environment, companies must develop and implement long-term strategies. For example, Fiberhome mitigated the negative impact of U.S. export controls by increasing its raw materials inventory, spending more on R&D, and actively expanding into overseas markets. Although these long-term countermeasures may enhance corporate resilience to a certain extent, they expose companies to greater financial pressures and risks. If the decision is wrong or the actual effect is not as expected, not only will they fail to enhance the resilience of the enterprise, but they may also cause more harm to the existing resources and operation of the enterprise, weakening its market competitiveness and sustainable development ability. Consequently, this study posits the subsequent research hypotheses.
Hypothesis 1. U.S. export controls on China reduce firm resilience.

## 2.2 Export controls, enterprise internationalization and enterprise resilience

With the rise of trade protectionism and the increase in trade barriers, export control policies have become an important means for the United States to protect its industries and markets, with far-reaching impacts on the market access and

supply chain layout of Chinese enterprises in the process of internationalization [1]. On the one hand, strict export control restricts Chinese enterprises' access to key technologies and raw materials. It excludes some of them from advanced technologies' global supply chain system. Such exclusion restricts Chinese enterprises' access to core technologies and high-end equipment. It severely limits their ability to integrate deeply into the global value chain and achieve resource sharing and complementarity. This may undoubtedly slow down the pace of Chinese companies' international expansion and technological innovation. The flexibility and adaptability of Chinese companies in global competition are being challenged. This phenomenon of "technological silos" puts Chinese enterprises in a disadvantageous position in international competition and makes it difficult for them to achieve technological leapfrogging and upgrading of the global value chain through international cooperation [39]. This is not only a restriction on the current market activities of enterprises but also a far-reaching impact on their future growth space and resilience building.

On the other hand, the scope of export control goes far beyond the export of hardware and equipment. It extends to the level of international exchanges and cooperation of technical personnel. Such restrictions hinder the deep integration of Chinese enterprises with advanced international technology and management experience and limit the enhancement of their competitiveness in the global market [40]. In addition, export control also increases the difficulty for Chinese enterprises in exploring overseas markets and reduces their import and export trade business [41,42]. Due to the inability to obtain the necessary equipment and technical support, the competitiveness of Chinese firms in the international market is weakened, and the implementation of their internationalization strategy is thus faced with challenges and uncertainties, further reducing the overall resilience of the firms. This is why the following study hypotheses are put forward in this paper.

Hypothesis 2. US export controls on China will reduce firm resilience by inhibiting firm internationalization.

## 2.3 Export controls, corporate finance and business resilience

Faced with the shackles of export controls, enterprises have had to face even more severe challenges in the area of financing. Tighter loan conditions, rising interest rates, and shorter loan terms have significantly reduced its financing capacity in a series of knock-on effects. Specifically, U.S. export control measures may limit Chinese enterprises' access to key technologies and equipment, affecting their normal production and R&D activities. These measures may lead to a decline in the competitiveness of Chinese firms in international markets, which in turn causes their ability to attract foreign investment and maintain existing debt financing [43]. At the same time, export controls can lead to higher operating costs and lower profits for Chinese firms [28]. Suppose certain controlled products or services cannot be imported from the United States through normal channels. In that case, Chinese firms may need to find alternative sources or develop the relevant technologies on their own, increasing the firms' procurement and R&D costs. These additional cost burdens may reduce the profitability of enterprises, which in turn may cause their ability to raise capital. In addition, economic sanctions may also trigger market uncertainty and affect FDI inflows [44]. It leads domestic and foreign investors to be more cautious. When the market is skeptical about a particular industry's prospects, the relevant firms' stock prices and investor confidence may take a hit. This change in the market sentiment may affect the financing environment of enterprises, making it more difficult and costly to raise funds, while financing restrictions will increase the financial burden of enterprises, and their market competitiveness and adaptability will be weakened, thus seriously weakening their resilience and viability [45]. Due to this, the subsequent research hypotheses are proposed in this paper.

Hypothesis 3. U.S. export controls on China will reduce firm resilience by shrinking the scale of corporate finance.

## 2.4 Export controls, enterprise risk, and enterprise resilience

The United States foreign export sanctions have resulted in a substantial increase in uncertainty in the international trade landscape, and the risks faced by enterprises have also risen. First, confronted with a significant decline in external demand, countries that previously relied on export diversification strategies had to adjust their export structure quickly and

drastically reduce exports to high-risk markets and products. This increased the volatility of export trade, directly under-mined the market foundation and stability of enterprises, and weakened their resilience. Moreover, strengthening export control will significantly increase the risk of Chinese enterprises in overseas operations [46,47]. It forces firms to increase compliance costs and additional information expenses [48]. It includes strengthening compliance regulations, building a comprehensive regulatory framework and trade compliance system, deepening risk assessment mechanisms, etc. These additional inputs aggravate firms' financial burden and increase the complexity and risks in the operation process, further weakening firms' risk-resistant ability and resilience. Further, from a macro perspective, if the U.S. export control policy is too harsh or lasts too long, it will have a far-reaching impact on China's economic development environment and finan-cial markets [49]. These impacts include, but are not limited to, a slowdown in economic growth [50], pressure on the job market [51], and instability in the financial markets [52]. The macroeconomic alterations will ultimately be conveyed to the firm level, intensifying the uncertainties and risks in the external environment, thus posing more severe challenges to the resilience and long-term development of firms. Based on this, the following hypotheses are proposed in this study. Hypothesis 4. U.S. export controls on China will reduce firm resilience by increasing firm risk.

## 3. Designment of research

### 3.1 Model measurement

This study investigates how U.S. export controls affect firm resilience by leveraging the release of the U.S. control list as a natural experiment-like scenario. It takes into account the multi-temporal gradual characteristics of control implementation. It refers to Beck [53] to construct a multitemporal double-difference model.

$$\text{Ressd}_{it} = \beta_0 + \beta_1 DID_{it} + \beta_2 Firm_{it} + \sigma_i + \theta_t + \varepsilon_{it} \tag{1}$$

The subscript i represents firms, and subscript t represents years. The explanatory variable $\text{Ressd}_{it}$ represents firm resilience. The core explanatory variable $DID_{it}$ means an export control dummy. $\beta_1$ is a double difference-in-differences estimator that measures the consequences of U.S. export controls on firm resilience. $Firm_{it}$ represents firm-level control variable. $\sigma_i$ represents individual firm fixed effects. $\theta_t$ captures time-invariant yearly effects, and $\varepsilon_{it}$ stands for the stochas-tic error component.

### 3.2 Indicator measurement and data description

**3.2.1 Explained variable.** Corporate resilience Re*ssd* serves as the key outcome variable in our analysis. The original meaning of resilience is recovery, rebound. Corporate resilience refers to the ability of firms to resist shocks, recover and rebound, and resume development under the impact of unfavorable events [54–56]. Most current studies use either operational or market-based indicators to measure the level of corporate resilience [57,58]. This paper refers to Ortiz-de-Mandojana and Bansa [22] and Pan et al. [59]. The main view is that corporate resilience is mainly reflected in the two aspects of high-performance growth and low financial volatility. The growth of enterprise's cumulative sales revenue in three years is used as an operating indicator to measure performance growth. Standard deviation of stock's monthly return serves as our financial volatility metric. These indicators are used to formulate corporate resilience indicator based on entropy method. Enterprise resilience, as the core ability of enterprises to resist risks, maintain operations, and achieve long-term growth in the global market, has been paid more and more attention by academia and industry.

**3.2.2 Explanatory variables.** The explanatory is the export control dummy variable *DID*. As a key export control instrument, the list of entities issued by the United States Government is complex and extensive. In terms of the logic of policy formulation, decisions on U.S. export control policies are mainly based on macro-strategic considerations such as U.S. national security and technological competition, rather than on the business conditions or resilience levels of

individual Chinese firms. Therefore, this paper adopts the U.S. export control policy as a quasi-natural experiment, which can avoid the potential reverse causality problem to a certain extent. Although directly targeting specific firms on the list, the scope of its impact goes far beyond that, which is demonstrating significant industry spillover effects. This spillover effect stems from the close connection of the global industrial chain and the sensitive reaction of market participants to control information. Even enterprises are not directly included in the list of entities. If they are within the controlled industries, they will also face the adverse impacts of declining market confidence, disruption of supply chains, and rising financing costs. Moving beyond immediate targets, this research examines the cascading effects of export restrictions on China's industrial landscape. In line with Yang and Zheng's [60] approach, we implement a dichotomous dummy variable to capture whether a company faces indirect consequences from its sector being subject to export controls. If a firm's industry is included in the export control category in a certain year, the treatment variable of the firm in the DID model takes the value of "1", indicating that it is indirectly affected by the policy from that year onwards and in subsequent years. Conversely, when a firm's industry is absent from the restricted list, the treatment variable is coded as "0" to denote no exposure to export controls.

**3.2.3 Control variables.** To address potential omitted variable bias in model estimation, this study incorporates multiple control variables. (1) Firm size (Size), calculated as the natural logarithm of total assets. Larger firms typically have stronger bargaining power, more stable supply chain networks, and better cash flows. It is more resilient to the negative impacts of export controls. Controlling Size excludes differences in firm resilience due to differences in resource endowments. It precises the identification of the independent effects of export controls. (2) Leverage (Lev), computed as total liabilities divided by total assets. High Lev may increase financial risk and limit the flexibility of firms to respond to crises. Export controls may lead to a deterioration in the financing environment. Controlling Lev distinguishes between the interaction effects of firms' own financial risk and policy shocks. (3) Age at listing (lnage), derived from the logarithmic transformation of years since IPO plus one. Companies that have been listed for a long time have usually accumulated richer management experience, stable customer relationships, and supply chain networks. lnage may affect firms' strategic choices to cope with external shocks. Controlling for this variable reduces estimation bias due to different stages of the firm's life cycle. (4) Return on Equity (ROE), measured as net profit relative to shareholders' equity. ROE reflects corporate profitability. High profitability may enhance firms' ability to buffer external shocks through internal funds. Controlling ROE precludes the impact of a firm's operating efficiency on resilience. It avoids the masking of the true effects of export controls by highly profitable firms that are well financed. (5) Growth, quantified via operating revenue growth rates. High-growth companies may rely more on external financing or international market expansion, which is more sensitive to export control shocks. Controlling growth distinguishes the heterogeneous effects of export controls on high-growth versus low-growth firms. (6) Current Ratio (Liqtratio), expressed as current assets divided by current liabilities. Liqtratio reflects short-term solvency. Insufficient liquidity may exacerbate financial distress from export control shocks. Controlling Liqtratio reduces estimation bias due to differences in an enterprise's short-term liquidity. (7) Cash holdings (Cash), represented by monetary assets and marketable securities relative to total assets. Cash holdings are important cushions for businesses against unexpected risks. Controlling cash holdings identifies whether the impact of export controls on firm resilience is independent of a firm's own risk-reserving strategy. (8) Price-earnings ratio (PE), calculated as share price divided by EPS. PE reflects the market's expectation of a company's future earnings. Highly valued companies may be at greater risk of share price volatility. Controlling PE reduces the disruption to resilience metrics caused by market sentiment swings. (9) Proportion of independent directors (Independ), this paper measures it by using the ratio of the number of independent directors to the total number of directors. The higher the Independ, the stronger the level of corporate governance usually is. It enhances the capacity for strategic corporate alignment. Controlling Independ can exclude interference with policy responses from internal corporate decision-making mechanisms. (10) Chairman's ownership of shares (Chair),

represented by the chairman's equity stake. Chair may incentivise a greater focus on long-term corporate resilience. Controlling Chair distinguishes the impact of export controls from differences in executive behavior. (11) Audited by top accounting firms (big4), measured by using a dummy variable. It takes a value of 1 if the firm is audited by one of the world's top four accounting firms, and 0 otherwise. (12) Nature of equity (SOE). Measured by using a dummy variable, which is 1 when the firm is a state-owned enterprise and 0 otherwise. State-owned enterprises may enjoy increased policy support and financing facilities. The nature of control equity identifies the differential impact of export controls on state-owned versus non-state-owned firms. (13) Herfindahl Index (HHI), measured by using the squared sum cumulative value of the ratio of the main business's operating income to the industry's total main business income. Industry concentration affects firms' ability to adapt to market changes. Controlling industry concentration reduces estimation bias due to differences in the competitive market environment.

Also, the article discusses excluding some potential factors. (1) Geographical location, although regional policies or resource endowments may affect firm resilience, this study has controlled for province-industry high-dimensional fixed effects. It can capture unobservables at the regional level. (2) Enterprise R&D investment. High R&D enterprises may influence enterprise resilience through technological innovation, but this study has controlled for ROE and Growth. It can identify the direct impact of export control on the level of enterprise internationalization, rather than exploring how enterprises indirectly alleviate the impact through innovative behaviors.

### 3.3 Data sources

The analysis draws upon an initial dataset comprising all manufacturing sector A-share listings from China's Shanghai and Shenzhen exchanges between 2015 and 2023. The dataset undergoes standard preprocessing procedures as follows. First, to establish a clean baseline, the dataset is purged of all firm years where companies carried ST, *ST, or PT status indicators during the research period. Such enterprises face concerns about risks or delisting procedures. Their financial behavior and anti-risk mechanisms may be significantly affected by non-market factors like regulatory intervention, and incentives to preserve the shell. This is fundamentally different from the resilience-forming mechanisms of healthy companies. Second, firms with short observation times may have more uncertainty and volatility. So the data of firms with less than three years of consecutive observations are excluded. Third, the dataset was purged of incomplete records pertaining to major variables to eliminate potential analytical distortions. Fourth, The analysis employed 1% symmetric Winsorization of firm-level continuous variables to reduce the undue influence of extreme observations. After this series of screening and processing, the final sample of 16,133 firm-year observations is obtained. The sample covers 29 broad industry categories in the manufacturing sector. It includes core areas such as electronic equipment manufacturing (C39), general equipment manufacturing (C34), and pharmaceutical manufacturing (C27). Nonetheless, data may still have some limitations. For example, selecting mainly A-share listed companies as the sample may lead to sample bias. Because these companies are relatively larger and have better governance structures. It may not represent all Chinese manufacturing enterprises, especially MSMEs. Due to the timeliness and availability of data, we may not be able to fully capture all the factors that may affect business resilience, which may lead to some bias. In addition, missing data may lead to selective bias in the sample. It has an impact on the generalisability and accuracy of research findings. In response to these potential data limitations, the paper will subsequently strengthen the robustness of the empirical findings through a series of robustness and endogeneity tests. Descriptive statistics for each variable in this paper are shown in Table 1.

The main sources of data involved in this paper are as follows (Table 1). (1) U.S. export control data are obtained from the official website of the U.S. BIS. The listed companies in China involved in the 2018–2023 entity list are manually screened. All firms were classified according to the standardized industry taxonomy provided in the CSRC's 2012 updated version of the Listed Company Industry Classification Guidelines. (2) The enterprise data mainly comes from CSMAR, WIND, and other authoritative databases.

**Table 1. Descriptive statistics of the main variables.**

|  | Count | Mean | Sd | Min | Max |
|---|---|---|---|---|---|
| Ressd | 16133 | 0.7620 | 0.0903 | 0.5963 | 0.9325 |
| DID | 16133 | 0.2425 | 0.4286 | 0.0000 | 1.0000 |
| Size | 16133 | 22.1132 | 1.1696 | 20.0145 | 25.7203 |
| Lev | 16133 | 0.3683 | 0.1769 | 0.0575 | 0.7851 |
| Inage | 16133 | 1.9316 | 0.9343 | 0.0000 | 3.3322 |
| ROE | 16133 | 0.0924 | 0.0657 | 0.0037 | 0.3511 |
| Growth | 16133 | 0.2049 | 0.3116 | −0.1853 | 1.7462 |
| Liqtratio | 16133 | 0.5966 | 0.1642 | 0.2081 | 0.9216 |
| Cash | 16133 | 0.2191 | 0.1439 | 0.0295 | 0.6863 |
| PE | 16133 | 76.9427 | 124.8794 | 6.1619 | 901.7677 |
| Independ | 16133 | 37.7379 | 5.3001 | 33.3300 | 57.1400 |
| Chair | 16133 | 11.7986 | 15.6219 | 0.0000 | 59.4500 |
| big4 | 16133 | 0.0501 | 0.2183 | 0.0000 | 1.0000 |
| SOE | 16133 | 0.2300 | 0.4208 | 0.0000 | 1.0000 |
| HHI | 16133 | 0.1571 | 0.1069 | 0.0412 | 0.6169 |

Data source: the original data of enterprise explanatory variables and control variables are obtained from the CSMAR database. Authors further measure them. Export control is compiled by the authors from the official website of BIS, the same below.

## 4. Empirical analysis

### 4.1 Benchmark regression analysis

Starting from a quasi-natural experiment, the U.S. control list release, the analysis applies a multi-temporal DID methodology to assess the effects of export regulations on firm-level resilience. The benchmark estimates are shown in Table 2. To ensure estimation stability, we conduct incremental regression analyses. To deal with potential problems, such as heteroskedasticity and autocorrelation, each model in Table 2 employs firm-level clustering for standard error estimation. The regression models presented in columns (1)-(3) sequentially introduce control variables followed by firm and year fixed effects (Table 2). The analysis reveals statistically significant negative coefficients ($p < 0.01$) for both the key explanatory

**Table 2. Benchmark regression results for export controls on firm resilience.**

|  | (1) | (2) | (3) |
|---|---|---|---|
|  | Ressd | Ressd | Ressd |
| DID | −0.0699*** | −0.0044*** | −0.0044*** |
|  | (0.0012) | (0.0013) | (0.0012) |
| Constant | 0.7789*** | 0.7630*** | 0.5808*** |
|  | (0.0008) | (0.0003) | (0.0286) |
| Control | No | Yes | Yes |
| Firm FE | No | No | Yes |
| Year FE | No | No | Yes |
| Observations | 16133 | 16133 | 16133 |
| Adjusted R-squared | 0.1100 | 0.8791 | 0.8849 |

Notes: Parentheses contain firm-level clustered robust standard errors. Asterisks *, **, and *** denote statistical significance at the 10%, 5%, and 1% levels respectively.

variables and export control measures, indicating that Entity List designations substantially reduce corporate resilience. Listed firms encounter immediate operational challenges including supply chain interruptions and competitive position deterioration, along with enduring consequences such as technology access limitations, market entry barriers, and global reputation deterioration. It seriously undermines the resilience of the enterprise. At the same time, this impact is not only limited to entities directly subject to export controls but also has indirect negative effects on a wider range of economic agents through complex supply chain networks and market psychological expectations. As a result, hypothesis 1 of the previous study has been verified.

## 4.2  Parallel trend test

Adopting the research design of Li et al. [61], we employ event study methodology to assess year-by-year variations in export control consequences for firm resilience. The following equation examines temporal heterogeneity in policy effects:

$$Ressd_{it} = \alpha_0 + \prod_{k \geq -6, k \neq 1}^{5} \delta_k D_{it_{it}}^k + \alpha_2 Firm_{it} + \sigma_i + \theta_t + \varepsilon_{it}$$

(2)

$D_{it}^k$ represents a dummy variable for the event of U.S. export controls, assuming that the year in which the U.S. export control lists are issued is T, such that k = year-T; when k = -6, -5,......, 4,5, the corresponding = 1, and 0 otherwise. k is 0 to denote the first year in which a firm is in an industry that is included on the control lists for export controls. The parameter k denotes event time, where k<0 indicates years preceding listing and k>0 denotes years following listing. The remaining variables align with Model (1)'s specification, using the pre-policy phase as the baseline to mitigate multicollinearity and establish temporal comparability. The time heterogeneity of the impact of export control list issuance can be assessed by comparing the statistical significance and size of the coefficient in equation (2).

Fig 1 presents the estimation results of $\delta_k$ (with 95 percent confidence intervals). It can be found that there is no significant difference in the resilience of firms in the treatment and control groups before the industry in which the firms are located is included in the list of export-controlled entities (Fig 1). The empirical evidence supports the satisfaction of the parallel pre-trends condition. In the year and the following year after the industry in which the located enterprises are listed, the estimated value of $\delta_k$ shows a significant negative change, and the negative effect increases year by year. This indicates that the U.S. export control measures have an immediate and significant negative impact on corporate resilience, and that this impact continues to increase in the short term. From the 2nd period after the policy, the estimates of $\delta_k$ for the treatment group and the control group recovered to the state of non-significant difference. This shows that companies may have responded to export controls by making a series of strategic adjustments, such as increasing investment in independent R&D, optimizing supply chain management, seeking alternative technologies and markets, and so on. These adjustments have alleviated direct or indirect negative pressure on companies caused by export controls, and have restored and enhanced their resilience. It also shows that the U.S.'s measures to curb the development of Chinese companies through export controls are not sustainable. It is similar to Liu and Li's [62] research conclusions.

## 4.3  Robustness test

**4.3.1  PSM-DID test.**  Consistent with Cassell et al. [63], we conduct robustness tests using a PSM-enhanced DID estimator to alleviate selection-related endogeneity and confirm our conclusions. To maintain inter-model comparability and ensure consistency in the causal identification logic of the two types of models with the underlying regression, this paper uses Size, Lev, lnage, ROE, Growth, Liqtratio, Cash, PE, Independ, Chair, big4, SOE and HHI as matching variables. We use them to run a logit regression to get the propensity score values. The cities with the closest propensity score values are the matched firms for firms in the export-controlled industry. This approach minimizes systematic differences in the level of organizational resilience of different firms. It reduces the bias of the double-difference model

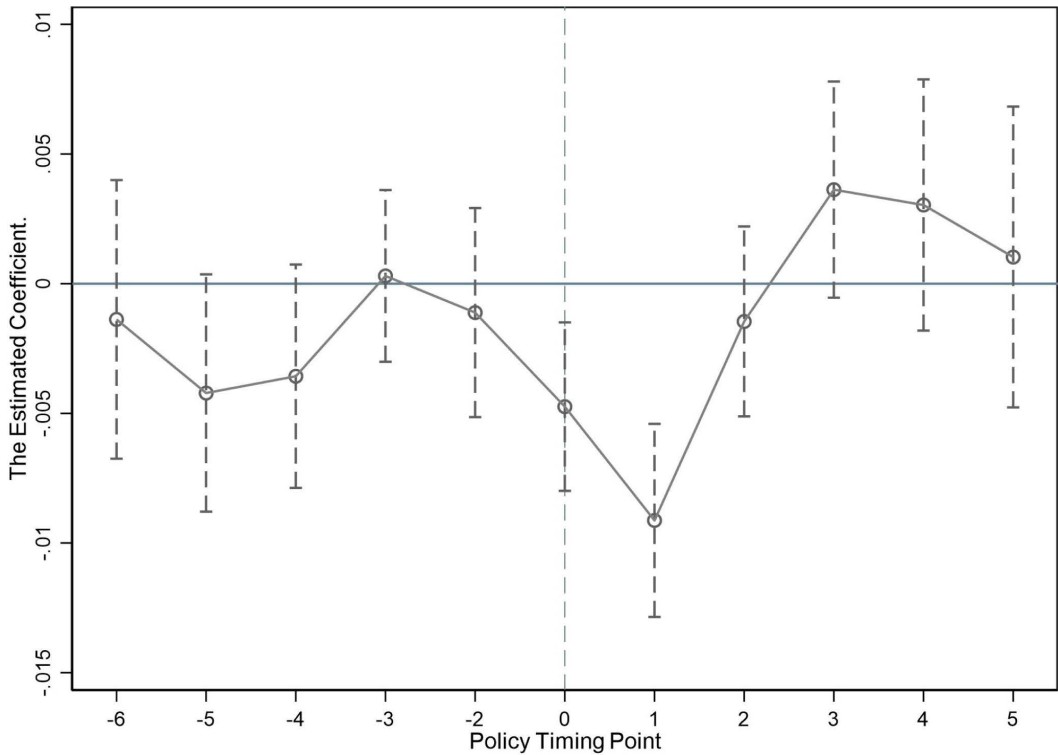

**Fig 1. Parallel trend test.**

estimation. Specifically, this paper uses nearest-neighbor 1:1 matching, radius matching, and kernel matching to match firms. The paper uses the matched samples for the base regression. Table 3 reports results. Consistent with our benchmark analysis, Table 3 documents uniformly negative and significant (at the 1% level) coefficients for the *DID* variable across all model variations.

**4.3.2 Placebo test.** In order to exclude the interference by other non-observed omitted variables such as other political, economic, and other policies, a placebo test is performed by randomly constructing a pseudo-experimental

**Table 3. Regression results based on PSM-DID.**

|  | (1) | (2) | (3) |
|---|---|---|---|
|  | **Nearest Neighbor Matching** | **Radius Matching** | **Core Matching** |
| DID | −0.0127*** | −0.0044*** | −0.0045*** |
|  | (0.0031) | (0.0012) | (0.0012) |
| Constant | 0.4757*** | 0.5791*** | 0.5818*** |
|  | (0.0577) | (0.0290) | (0.0287) |
| Control | Yes | Yes | Yes |
| Firm FE | Yes | Yes | Yes |
| Year FE | Yes | Yes | Yes |
| Observations | 5057 | 15845 | 16097 |
| Adjusted R-squared | 0.8678 | 0.8853 | 0.8851 |

group. It is used to make sure that resilience weakening effects by U.S. export controls are correct. Referencing Bradley et al. [64], a placebo test was carried out by randomly allocating experimental groups. Specifically, the experimental groups were randomly assigned through 500 random bootstrap samplings. It randomly selects firms as pseudo-regulated firms and repeats the estimation of the 'pseudo-effect' of export controls. Fig 2 displays the kernel density distribution of placebo test coefficients alongside their corresponding p-value scatterplot. The coefficient estimates from randomized treatment assignments concentrate near zero, with the majority failing to reach statistical significance even at the 10% level (Fig 2). Notably, our baseline estimates fall outside this null distribution, indicating that export controls' negative impact on firm resilience remains robust to potential omitted variable bias. Placebo test results support the conclusions of the baseline regression. Results strengthen the chain of causal inference between export controls and firm resilience. It means that the reduction in firm resilience is indeed driven by the export controls themselves rather than other unobserved confounders.

We assess the time-sensitivity of our results by conducting counterfactual analyses that predate the Entity List designations by both 2-year and 3-year intervals. It conducts a placebo test using a dummy for the year of policy occurrence instead of the policy's year of policy occurrence, and none of the results are significant (S2 Table). It suggests that the conclusions of this paper are still robust after accounting for the non-randomness of the time point of the policy shock. It excludes the possibility that 'differences in the resilience of firms in the experimental and control groups are dominated by time trends'. This further supports the core finding of a causal relationship between export control shocks and changes in firm resilience.

**4.3.3 Other robustness tests.** First, add other control variables. Enterprise innovation capacity and government subsidies may affect the level of enterprise resilience. To avoid estimation errors caused by omitted variables, this

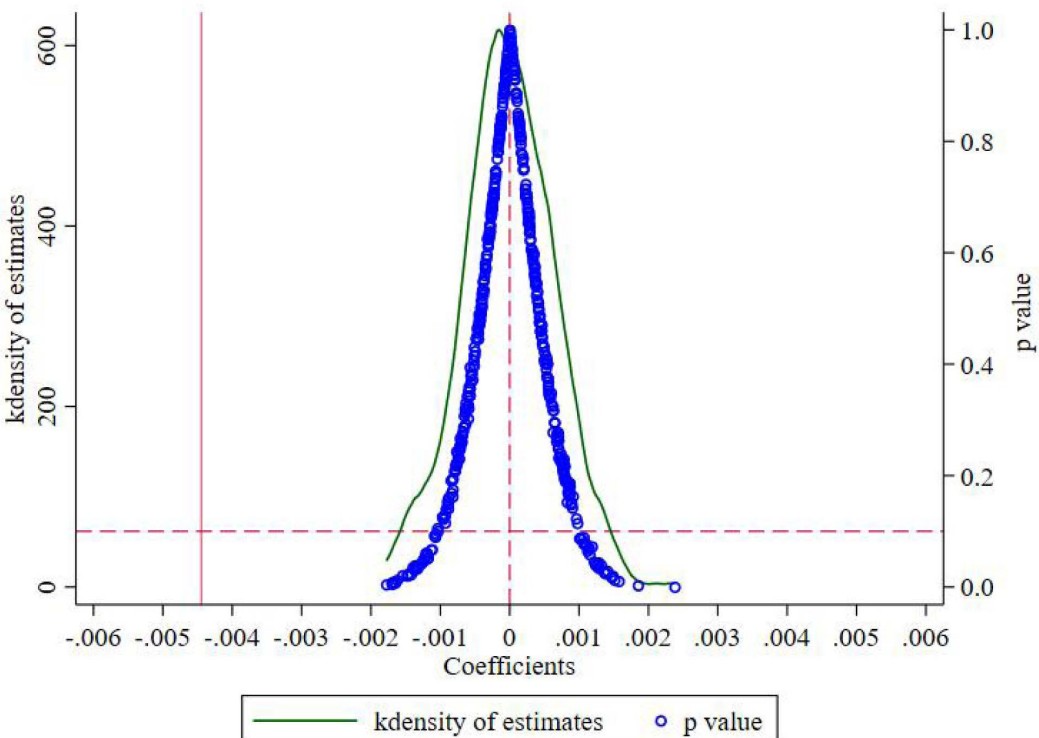

**Fig 2. Individual Placebo test.**

paper includes the variables of enterprise invention patents and government subsidies in the regression equation. Firm innovation capacity is measured by the logarithm of the number of invention patents granted by the firm plus 1. Government grants are measured by the ratio of the amount of government grants received by the enterprise to the enterprise's total assets. As evidenced by Table 4's first column results, the coefficient estimates align with our benchmark analysis, confirming the robustness of our primary conclusions.

Secondly, add province-industry high-dimensional fixed effects. Given that there may still be some industry characteristics that vary across provinces that may affect the level of firm resilience, which in turn may lead to estimation bias, the analysis additionally incorporates province-by-industry interaction fixed effects. In the regression results in column (2) of Table 4, after adding multiple fixed effects, the significance level of export controls is the same as that of the baseline regression. This shows that the conclusions are not affected by multiple fixed effects. The model is relatively robust.

Thirdly, adjust the clustering standard errors. Considering that some variables may be correlated in higher dimensions, this paper adjusts the standard error of clustering from the enterprise level to the industry level. The regression results are shown in column (3) of Table 4, which are not significantly different from the previous conclusions.

Fourthly, lag the control variables. Addressing possible simultaneity bias, we re-estimate the model using lagged controls. The results (Table 4, column 4) confirm the robustness of our key finding, with the policy variable remaining statistically significant (1% level) and positively signed.

## 5. Further analysis

### 5.1 Analysis of impact mechanisms

The theoretical analysis in the previous section shows that export control has reduced the level of firm resilience by inhibiting firm internationalization, shrinking the scale of financing, and exacerbating firm risk. This section tests these three mechanisms to clarify the specific conduction path of the export control affecting firm resilience. Based on equation (1), we set up the model for mechanism analysis as follows:

**Table 4. Other robustness tests.**

|  | (1) | (2) | (3) | (4) |
|---|---|---|---|---|
|  | Adding Control Variables | Adding High-dimensional Interaction Items | Changing the Clustering Criteria Error | Control Variables Lagged by One Period |
| DID | −0.0046*** | −0.0045*** | −0.0044* | −0.0041*** |
|  | (0.0012) | (0.0012) | (0.0024) | (0.0012) |
| Patent | −0.0006 |  |  |  |
|  | (0.0005) |  |  |  |
| Subsidy | −0.0411 |  |  |  |
|  | (0.0958) |  |  |  |
| Constant | 0.5877*** | 0.5923*** | 0.5808*** | 0.5404*** |
|  | (0.0322) | (0.0305) | (0.0332) | (0.0288) |
| Control | Yes | Yes | Yes | Yes |
| Firm FE | Yes | Yes | Yes | Yes |
| Year FE | Yes | Yes | Yes | Yes |
| Sicda#PROVINCECODE_13 | No | Yes | No | No |
| Observations | 14366 | 16128 | 16133 | 11731 |
| Adjusted R-squared | 0.8684 | 0.8813 | 0.8849 | 0.9139 |

Data source: Original data on firms' innovation capacity and firms' government grants are from the CSMAR database. Authors further measure them.

$$\text{Re}ssd_{it} = \beta_0 + \beta_1 DID_{it} + \alpha_1 DID_{it} \times M + \alpha_2 M + \beta_2 Firm_{it} + \sigma_i + \theta_t + \varepsilon_{it} \tag{3}$$

Where M is a mechanism variable, specifically a proxy variable for firm internationalization, firm financing, and firm risk respectively, and the other symbols have the same meaning as in equation (1).

**5.1.1 Inhibitory effects of firm internationalization.** This paper draws on the methods of Pisani et al. [65], and Sun et al. [66]. Breadth of corporate internationalisation structure (Overfirm) and intensity of firms' internationalization (Oversale) are used to measure the level of internationalization, respectively. This is measured by the breadth of the firm's internationalization structure in terms of the number of overseas subsidiaries as a proportion of the firm's total number of subsidiaries. This indicator reflects the strategic depth of a company's ability to internationalize through the deployment of overseas branches. Higher values indicate better cross-border network construction. The intensity of a firm's internationalization is measured by the ratio of the firm's overseas operating revenue to its total operating revenue. Higher values indicate a higher intensity of internationalization. Based on formula (1), this paper cross-processes the mechanism variables of enterprise internationalization with the policy variable DID, and brings the cross-term and its separate term into the main regression (formula 3). The regression results are shown in columns (1)-(2) of Table 5. The coefficient of

**Table 5. Analysis of the impact mechanisms: the dampening effect of firm internationalization and the contraction effect of the scale of firm financing.**

|  | (1) | (2) | (3) | (4) |
|---|---|---|---|---|
|  | Ressd | Ressd | Ressd | Ressd |
| DID×Overfirm | −0.0137* |  |  |  |
|  | (0.0070) |  |  |  |
| DID | −0.0035*** | −0.0031** | −0.0007 | −0.0026 |
|  | (0.0012) | (0.0014) | (0.0025) | (0.0029) |
| Overfirm | 0.0070 |  |  |  |
|  | (0.0049) |  |  |  |
| DID×Oversale |  | −0.0071** |  |  |
|  |  | (0.0035) |  |  |
| Oversale |  | 0.0030 |  |  |
|  |  | (0.0042) |  |  |
| DID×Finasset |  |  | −0.0446*** |  |
|  |  |  | (0.0151) |  |
| Finasset |  |  | −0.0263*** |  |
|  |  |  | (0.0064) |  |
| DID×banklev |  |  |  | −0.0210* |
|  |  |  |  | (0.0117) |
| banklev |  |  |  | −0.0115 |
|  |  |  |  | (0.0092) |
| Constant | 0.5800*** | 0.5778*** | 0.6236*** | 0.6128*** |
|  | (0.0287) | (0.0288) | (0.0566) | (0.0437) |
| Control | Yes | Yes | Yes | Yes |
| Firm FE | Yes | Yes | Yes | Yes |
| Year FE | Yes | Yes | Yes | Yes |
| Observations | 16133 | 16133 | 4612 | 8005 |
| Adjusted R-squared | 0.8849 | 0.8849 | 0.8956 | 0.9051 |

Data Source: Original data for the internationalization of enterprises variable from the Wind database. Authors further measures them. Raw data on the size of corporate finance are from the CSMAR database. Authors further measures them.

the cross-term $\alpha_1$ is significantly negative (Table 5). It shows that the resilience of firms with lower internationalization decreases more significantly than firms with higher internationalization, i.e., the export control reduces the resilience of firms by inhibiting firms' level of internationalization. This may be because export controls affect firms' internationalization by restricting their market access, hindering the integration of their global supply chains, and reducing opportunities for technological exchanges and cooperation. Such restrictions reduce firms' market diversification and their ability to cope with external shocks, further weakening their economic resilience and adaptability. As a result, the mechanism by which export controls weaken firms' resilience by inhibiting firms' internationalization.

**5.1.2 Contractionary effects of the scale of corporate finance.** Following Cheng et al. [67], we measure corporate external financing using two indicators: (1) Finasset – total financing inflows (debt + equity + related-party financing) scaled by total assets, and (2) banklev – the ratio of bank borrowings (long- and short-term) to total assets. The corresponding regression outputs appear in columns (3)-(4) of Table 5. The coefficient of the cross-multiplier $\alpha_1$ is negative. It indicates that firms with constrained corporate finance are more resilient than firms with better financing, i.e., export controls reduce firm resilience by shrinking the size of corporate finance. This may be because export controls raise firms' financing constraints in the capital market by increasing their operational uncertainty and compliance costs. The shortage of capital will weaken firms' ability to deal with emergencies in the face of unforeseen events, reduce their flexibility and room for strategic adjustment in the event of economic recession or sudden changes in market demand, and thus have a significant negative impact on firm resilience. As a result, the research hypothesis 3 of this paper, the mechanism of export control acting on enterprise resilience through the channel of financing scale contraction, can be supported and verified.

**5.1.3 Exacerbating effects of enterprise risk.** Building on prior work by Kothari et al. [68] and Hutton et al. [69], we quantify stock price crash risk through negative conditional skewness (Ncskew) of market-adjusted weekly returns. For downside risk measurement, we follow Miller and Leiblein [70] and He et al. [71], computing annual ROA deviations from target levels using the specified formula:

$$Downsiderisk_{it} = \sqrt{\frac{1}{5}\sum_{t=1}^{5}\left(ROA_{i,t-1} - iROA_{i,t-1}\right)^2}$$

(4)

$ROA_{i,t-1}$ denotes the actual annual ROA of firm I in year t-1. $iROA_{i,t-1}$ denotes the average annual ROA of the industry in which firm i operates in year t-1, which is the target value. The firm risk profile is measured by firm stock crash risk and firm downside risk, respectively. Results are shown in columns (1)-(2) of Table 6. The coefficient of the cross-multiplier term is significantly positive (Table 6). It indicates that firms with higher firm risk are more resilient relative to firms with lower firm risk. Export control reduces the resilience of firms by exacerbating firm risk. This may be because export controls increase the risk of supply chain disruptions and may cause the market to question the reliability of the products and services of the regulated firms, raising concerns among financial institutions and investors about the financial condition of the regulated firms, leading to a decline in market share and exacerbating market risk. The increase in enterprise risk will make financial stability, leading to a decline in business confidence, reducing the enterprise innovation ability and organizational effectiveness, and ultimately hurting the resilience of enterprises. Thus, this paper's research hypothesis 4 export control through the channel of risk exacerbation on the mechanism of enterprise resilience can be supported and verified.

## 5.2 Heterogeneity analysis

Existing literature suggests that there is significant heterogeneity in characterizing firm resilience, so is there also heterogeneity in the enhancement of export controls on firm resilience? This part provides an in-depth study of this to explore the negative effects of export controls further.

**5.2.1 Heterogeneity analysis based on competitive advantage in product markets.** Competitive advantage in the product market implies that a firm has a unique position in the market. It is an important factor. It affects firm resilience. In this paper, we refer to Frésard [72] and Yang et al. [73], which used the annual, industry median-adjusted growth

**Table 6. Analysis of impact mechanisms: exacerbating effects of enterprise risk.**

| | (1) | (2) |
|---|---|---|
| | Ressd | Ressd |
| DID×Downsiderisk | 0.1111*** | |
| | (0.0337) | |
| DID | −0.0090*** | −0.0035*** |
| | (0.0017) | (0.0011) |
| Downsiderisk | −0.0842*** | |
| | (0.0159) | |
| DID×Ncskew | | 0.0019** |
| | | (0.0009) |
| Ncskew | | 0.0056*** |
| | | (0.0005) |
| Constant | 0.6186*** | 0.5884*** |
| | (0.0406) | (0.0282) |
| Control | Yes | Yes |
| Firm FE | Yes | Yes |
| Year FE | Yes | Yes |
| Observations | 8945 | 15110 |
| Adjusted R-squared | 0.9081 | 0.9059 |

Data Source: Raw datas on corporate control variables are from the CSMAR database. Authors further measures them. Export controls compiled by authors from the official BIS website. Firm stock crash risk variables and firm downside risk variables are from the CSMAR database.

rate of main business revenue to measure the product market competitive advantage of enterprises. This indicator can reflect the change in the market share of the enterprise relative to competitors in the industry. A higher numerical value corresponds to stronger competitive positioning in the product market. Enterprises are divided into two groups of higher and lower product market competitive advantage according to whether they are above the annual median of the indicator. Results are reported in rows (1)-(2) of Table 7. It finds that the coefficient of core explanatory variable and

**Table 7. Heterogeneity analysis: heterogeneity based on competitive advantages and analyst focus in product markets.**

| | (1) | (2) | (3) | (4) |
|---|---|---|---|---|
| | Ressd | Ressd | Ressd | Ressd |
| | High Competitive Advantage in the Product Market | Low Competitive Advantage in Product Market | High Analyst Attention | Low Analyst Attention |
| DID | −0.0010 | −0.0093*** | −0.0023 | −0.0082*** |
| | (0.0019) | (0.0017) | (0.0015) | (0.0028) |
| Constant | 0.5785*** | 0.5816*** | 0.5690*** | 0.5733*** |
| | (0.0433) | (0.0414) | (0.0378) | (0.0703) |
| Control | Yes | Yes | Yes | Yes |
| Firm FE | Yes | Yes | Yes | Yes |
| Year FE | Yes | Yes | Yes | Yes |
| Observations | 7827 | 7861 | 10610 | 4427 |
| Adjusted R-squared | 0.8823 | 0.8918 | 0.8934 | 0.8731 |

Data source: raw datas on product market competitive advantage and analyst attention from CSMAR database. Authors further measures them.

export control is significantly negative in the lower product market competitive advantage group (Table 7). In comparison, it is not significant in the group with a higher product market competitive advantage. This shows that firms with a higher competitive advantage in product markets may be better able to withstand the negative impact of export controls on their business. Firms with a higher competitive advantage in product markets often have stronger market diversification capabilities, can expand their markets globally, and are less dependent on the U.S. market, thereby reducing the negative impact of export controls. Enterprises with lower competitive advantages in product markets have weaker market diversification capabilities. Export controls' impact on market competitiveness is more significant.

**5.2.2 Heterogeneity analysis based on analyst attention.** The degree of analyst attention is of great significance to enterprises, as it not only affects the information environment, capital cost, and corporate governance of enterprises, but also their market reputation, risk management capabilities, and strategic decision-making, thereby affecting their resilience. Positive analyst ratings can stabilize investor sentiment, reduce share price volatility, and ease financing constraints. Under export control constraints, market reputation is key for firms to maintain cash flow and invest in alternative technologies. Continuous tracking of corporate strategy by analysts can force management to optimize the risk management framework. Building on Zhou et al. [74], we operationalize analyst attention through the logarithmic transformation (base e) of the count of analysts providing earnings estimates per firm year, incremented by one. Firms are classified into two groups of higher and lower analyst attention according to whether they are above the annual median of this indicator, and the group regression results are reported in rows (3)-(4) of Table 7. The results show that the coefficient of the core explanatory variable export control is significantly negative in the group with lower analyst attention, while it is not significant in the group with higher analyst attention. This may be because analysts' attention brings market recognition and investor trust to the firms by increasing the transparency and accessibility of firms' information, and also leads to greater emphasis on the construction of internal control and risk assessment mechanisms, which improves the ability to identify, prevent, and respond to potential market risks through regular risk assessment and strategic planning. On the other hand, enterprises with low analyst attention lack sensitivity and responsiveness to external risks, making it difficult for them to identify and respond promptly to challenges posed by policy changes such as export controls, thus reducing their market adaptability and corporate resilience.

**5.2.3 Heterogeneity analysis based on political affiliation.** Political connections are often seen as a channel that can help companies obtain key resources and alleviate policy restrictions. They can provide companies with more policy support and resource guarantees, such as tax incentives and financing facilities. Through political affiliation, firms may be able to gain earlier access to information on policy changes. Enterprises can adjust their strategies in advance to mitigate the negative impact of export controls. Firms may be able to access scarce resources through government channels, enhancing their ability to cope with policy shocks. This facilitates communication between business and government, giving them more flexibility in policy implementation. These will have some positive impact on firms' response to export controls. Political affiliation is often defined as an implicit link between business and government. The government background of executives is widely used as a measure of the strength of a firm's political affiliation. This paper refers to the research of Luo and Liu [75]. The article measures the strength of firms' political affiliation by using whether a firm's manager has a track record as a government official. Executives with a government background may have easier access to policy support, financing facilities, and market access opportunities. This increases the resilience of firms in the face of external policy shocks. In concrete terms, political connections are scored as 1 if either the chairman or the general manager of a company is or has been a government official, and as 0 otherwise. According to the different political affiliations, this paper classifies companies into two groups, politically connected and unconnected. Table 8, rows (1)-(2), reports the regression results for the two groups. The results show that the coefficient of the core explanatory variable, the export control, is significantly negative in the no-connections group, while insignificant in the connected group (Table 8). This may be because firms without political affiliation are more vulnerable to external policy shocks and have difficulty in responding effectively through internal adjustment or acquisition of external resources, which has a significant negative

**Table 8. Heterogeneity analysis: heterogeneity based on political affiliation and corporate strategic aggressiveness.**

| | (1) | (2) | (3) | (4) |
|---|---|---|---|---|
| | Ressd | Ressd | Ressd | Ressd |
| | Politically Motivated Related Parties | Apolitical Related Parties | High Corporate Strategy Aggressiveness | Low Corporate Strategy Aggressiveness |
| DID | −0.0041 | −0.0056*** | −0.0045 | −0.0056*** |
| | (0.0025) | (0.0016) | (0.0054) | (0.0014) |
| Constant | 0.6695*** | 0.6130*** | 0.6424*** | 0.5838*** |
| | (0.0624) | (0.0420) | (0.1135) | (0.0420) |
| Control | Yes | Yes | Yes | Yes |
| Firm FE | Yes | Yes | Yes | Yes |
| Year FE | Yes | Yes | Yes | Yes |
| Observations | 2982 | 9011 | 520 | 9093 |
| Adjusted R-squared | 0.8557 | 0.8290 | 0.9211 | 0.9079 |

Data source: the original data on political affiliation and corporate strategic aggressiveness were obtained from the CSMAR database. Authors further measure them.

impact on firm resilience. In the group with political affiliation, the effect of export control policies on firm resilience is insignificant, suggesting that political affiliation provides an effective buffer mechanism for firms to mitigate or avoid the negative impacts of policy shocks and that such a protective effect may stem from the advantages of political affiliation in terms of policy information, smoother channels of communication with the government, and possible policy support and easy access to resources.

**5.2.4 Heterogeneity analysis based on the aggressiveness of corporate strategies.** Strategic aggressiveness as a corporation's manifestation of strategic behavior reveals the extent to which a company is proactive, risk-taking, and innovative in its strategic planning and development. Firms with a high degree of strategic aggressiveness are usually more innovative and flexible. They are able to quickly adjust resource allocation and market strategies in the face of external shocks. Firms tend to have greater capacity to allocate resources. They find alternatives within policy constraints. They mitigate the negative impact of policy shocks through diversification and forward planning. Thereby they increase the enterprise's resilience to risk. This paper refers to Sun et al. [76], Meng et al. [77] and Wang et al. [78]. The calculation method is to score each indicator from the dimensions of the tendency of innovation (measured by intangible asset), the tendency to expand the market (measured by the sum of selling and administrative expenses as a percentage of operating income), the growth potential (measured by the growth rate of operating income), the production efficiency (measured by the ratio of number of employees to revenue), the stability of organizational structure (measured by the ratio of the standard deviation of the number of employees over five years to the mean of the number of employees) and the capital intensity (measured by the ratio of fixed assets to total assets). We use them to obtain a score for the degree of aggressiveness of the corporate strategy. The higher the score, the more aggressive the strategy. A score of 0–17 indicates a non-aggressive strategy, while a score of 18–24 indicates an aggressive strategy. The sample is partitioned into two subsets according to strategic aggressiveness levels (high/low). These groups' regression outcomes are documented in rows (3)-(4) of Table 8. The results show that the coefficient of the core explanatory variable export control is significantly negative in the low strategic aggressiveness group, while it is not significant in the high strategic aggressiveness group. A low degree of strategic aggressiveness implies that firms may adopt more conservative strategies in the resource allocation, product development, and market expansion. In the face of U.S. export controls, such firms may lack the flexibility and foresight to adjust their resource allocation quickly, leading to lower resilience. In

opposition to their counterparts, highly strategically aggressive firms characteristically possess stronger inventive potential and adaptive flexibility, allowing more efficient mitigation of exogenous disturbances.

**5.2.5 Heterogeneity analysis based on supply chain concentration.** The supply chain is the 'bloodline' of business operations. Its structure and management mode of supply chain directly determines the survivability of enterprises in the complex environment. It directly determines the enterprise's ability to adapt and recover in the face of external shocks. According to the theory of resource dependence and dynamic capacity theory, higher supply chain concentration means that enterprises have a higher degree of dependence on a small number of core suppliers or customers. This dependence may exacerbate the vulnerability of enterprises in export control situations.

In this paper, the customer concentration Herfindahl index and the supplier concentration Herfindahl index are used to measure supply chain concentration, respectively. According to whether they are higher than the median of customer concentration or supplier concentration, the samples are divided into two groups high customer concentration or high supplier concentration and low customer concentration or low supplier concentration. Group regressions are carried out. As shown in columns (1)-(4) of Table 9, the coefficients of the export control in the customer (supplier) concentration are significantly negative in the group with high concentration. It is not significant in the group with low. This may be because of the fact that firms with high usually have deep dependence on a few key counterparties. If core suppliers are unable to supply due to controls like key components being restricted. When U.S. export control policies are implemented, it is difficult to find alternative sources in the short term. It leads to production stagnation and order delays, which directly affects the resilience of firms' operations. If major customers are located in regulated markets such as the United States or its allies. The enterprise may face a loss of orders. High concentration means that it is difficult to shift markets in the short term and a sharp drop in revenue exacerbates financial vulnerability.

**5.2.6 Heterogeneity analysis based on environmental uncertainty.** Uncertainty in the economic environment may increase the risk and uncertainty of firms in the international market, thus affecting firm resilience. Following Baker et al. [79], we employ their news-based Economic Policy Uncertainty (EPU) index, derived from textual analysis of the South China Morning Post. Our annualized measure averages 12 monthly indices (divided by 100 for scaling). Stratifying the sample by median EPU values into high/low uncertainty groups, we find significantly negative export control coefficients only in high-uncertainty environments (Table 9, rows 5–6). This aligns with theory, as elevated uncertainty amplifies market volatility, demand fluctuations, and supply chain risks. At this time, the imposition of U.S. export sanctions can

**Table 9. Heterogeneity analysis: heterogeneity based on supply chain concentration and environmental uncertainty.**

|  | (1) | (2) | (3) | (4) | (5) | (6) |
|---|---|---|---|---|---|---|
|  | Ressd | Ressd | Ressd | Ressd | Ressd | Ressd |
|  | High Customer Concentration | Low Customer Concentration | High Supplier Concentration | Low Supplier Concentration | High Environmental Uncertainty | Low Environmental Uncertainty |
| DID | −0.0058*** | −0.0021 | −0.0057*** | −0.0012 | −0.0070*** | −0.0010 |
|  | (0.0016) | (0.0021) | (0.0015) | (0.0024) | (0.0020) | (0.0027) |
| Constant | 0.5555*** | 0.5835*** | 0.5281*** | 0.6550*** | 0.3101*** | 0.7524*** |
|  | (0.0353) | (0.0672) | (0.0547) | (0.0501) | (0.0711) | (0.0522) |
| Control | Yes | Yes | Yes | Yes | Yes | Yes |
| Firm FE | Yes | Yes | Yes | Yes | Yes | Yes |
| Year FE | Yes | Yes | Yes | Yes | Yes | Yes |
| Observations | 10,004 | 5,632 | 10,074 | 5,495 | 9651 | 6139 |
| Adjusted R-squared | 0.8768 | 0.8939 | 0.8796 | 0.8970 | 0.7648 | 0.9039 |

Data sources: Customer concentration and supplier concentration are from the CSMAR database; the original data on environmental uncertainty are from the China Economic Policy Uncertainty Index and are further measured by the authors.

further exacerbate these uncertainties, leading to greater operational pressures on firms. In periods of lower economic uncertainty, firms usually have more flexibility and resources to cope with external shocks, making it difficult to respond effectively to the negative impact of sanctions.

## 6. Conclusions and policy recommendations

This study compiles comprehensive data from the U.S. BIS Entity List and examines financial/operational metrics for Chinese A-share manufacturing firms (2015–2023) to analyze the multifaceted effects of U.S. export controls on corporate resilience. The findings show that U.S. export control measures significantly weaken the resilience level of Chinese firms in the affected industries, revealing the profound impact of international trade policy changes on firms' long-term adaptive and resilience capabilities. Furthermore, this paper reveals three major transmission paths of this negative effect through the mechanism analysis. Firstly, the export control restricts the internationalization process of firms, which leads to the obstruction of overseas market expansion. Secondly, the deterioration of the financing environment and the contraction of financing channels exacerbate the pressure on firms' capital. Lastly, the uncertainty and potential risk of trade policy increases, forcing firms to face higher operating costs and market risks. At the level of extended analysis, this paper finds that Chinese firms with higher competitive advantages in product markets, more attention from analysts, and more aggressive development strategies show relative resilience in the face of U.S. export controls and can effectively buffer against external shocks. On the contrary, firms that lack political affiliation or are in a high degree of uncertainty in the external environment are more vulnerable to negative shocks from export control policies, and their firm resilience is significantly reduced.

We list a few limitations to this study. First, the study is based on a sample of Chinese A-share listed manufacturing companies and does not include unlisted companies and SMEs. Given the significant differences in SMEs' access to resources, supply chain integration and policy responsiveness, the impact of export controls on SMEs is likely to be more complex. Second, the study data mainly come from the publicly available BIS entity list and CSMAR database, but key variables such as the specific structure of firms' supply chains, undisclosed government subsidies or informal political connections are difficult to obtain. Third, although the empirical data are available until 2023, the dynamically evolving nature of export controls requires continuous monitoring of their impact over time, especially the lagged effects of multiple rounds of technology embargo policies on firms.

In future, research may be expanded in two ways. On the one hand, the mechanism of China's countermeasures on firms' resilience can be explored in depth, analysing how policies such as tariff countermeasures and technology reciprocity controls affect firms' adaptive capacity and innovation reconfiguration paths in the export control environment. On the other hand, case studies can be used to supplement the lack of quantitative data, focusing on the differential impact of undisclosed micro characteristics of firms on export control responses. Based on the empirical findings of this study, we propose strategic recommendations to enhance corporate resilience against trade policy turbulence.

Firstly, the diversification of supply chains should be promoted and international cooperation deepened. To enhance the resilience of enterprises to external shocks, enterprises should be encouraged to expand their areas of cooperation with non-sanctioned countries, build a diversified supply chain system, and reduce their reliance on a single source of supply, thereby enhancing the resilience and security of the supply chain. At the same time, promoting the establishment of multilateral export control coordination mechanisms can mitigate the potential impact of the policies of a single country on the global supply chain and promote international cooperation and coordination. In addition, enterprises should actively participate in international cooperation to significantly enhance their international competitiveness through technological exchanges and market cooperation.

Secondly, the Government's support and financial guarantee system should be improved. Governments play an important role in addressing the challenges posed by export controls. To face challenges, they should enhance financial support for affected enterprises, and provide preferential loans, credit guarantees, and other financing facilities. They also should optimize the capital market environment and reduce the cost of enterprise financing. In addition, enterprises should

be encouraged to establish a close communication mechanism with government departments to keep abreast of policy trends and formulate more precise and effective coping strategies to enhance their resilience and competitiveness in the complex and changing external environment.

Thirdly, risk management and enterprise strategic planning should be strengthened. Organizations must capitalize on modern technological tools (such as large-scale data processing, artificial cognition, and decentralized verification mechanisms) to formalize risk governance structures and amplify their competence in discerning, overseeing, and addressing value chain and market exposures. At the same time, enterprises are encouraged to cooperate with third-party risk assessment organizations to improve the accuracy and responsiveness of risk assessment.

Fourthly, technological innovation and high-quality development strategies should be promoted. The government should increase its support for high-tech enterprises and research and development activities, and promote the enhancement of the independent innovation capacity of enterprises. Cooperation among industries, universities, and research institutes, should accelerate the transformation of scientific and technological achievements and enhance the market competitiveness of enterprises. Enterprises are encouraged to establish close partnerships with universities and research institutes to carry out technological research and development and talent training jointly. Enterprises should take technological innovation as a key way to enhance their core competitiveness and through continuous investment in R&D to achieve sustainable development.

Fifthly, the information disclosure and market communication mechanisms should be enhanced. Transparency is an important factor in enhancing the market recognition and financing ability of enterprises. Enterprises should take the initiative to improve the quality of information disclosure, strengthen communication with market players such as analysts and investors, and convey timely information on their operations and developments to enhance market confidence.

Sixthly, flexible strategies and internal management optimization should be strengthened. In the rapidly changing market environment, enterprises need to adopt flexible and forward-looking strategies to improve their adaptability to market changes. Strengthening internal management and team building to enhance the enterprise's strategic execution and decision-making efficiency ensures that the enterprise can respond quickly to market changes and achieve its strategic objectives.

## Supporting information

**S1 Fig. The number of Chinese subject to export enterprises control.**
(PDF)

**S2 Table. Time placebo test.**
(PDF)

**S3 File. Specific case of U.S. tightens tech restrictions fiberhome placed on BIS entity list.**
(PDF)

## Author contributions

**Conceptualization:** Huan He, Xianjing Huang.

**Data curation:** Huan He, Xianjing Huang.

**Formal analysis:** Huan He, Xianjing Huang.

**Funding acquisition:** Huan He, Xianjing Huang.

**Investigation:** Huan He, Xianjing Huang.

**Methodology:** Huan He, Xianjing Huang.

**Project administration:** Huan He, Xianjing Huang.

**Resources:** Huan He, Xianjing Huang.

**Software:** Huan He, Xianjing Huang.

**Supervision:** Huan He, Xianjing Huang.

**Validation:** Huan He, Xianjing Huang.

**Visualization:** Huan He, Xianjing Huang.

**Writing – original draft:** Huan He, Peng Chen, Xianjing Huang.

**Writing – review & editing:** Huan He, Peng Chen, Xianjing Huang, Le Li.

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
