## [Decision Letter · Decision Letter 0]

27 Feb 2025

Dear Dr. Huang,

Thank you for submitting your manuscript to PLOS ONE. After careful consideration, we feel that it has merit but does not fully meet PLOS ONE’s publication criteria as it currently stands. Therefore, we invite you to submit a revised version of the manuscript that addresses the points raised during the review process.

We look forward to receiving your revised manuscript.

Kind regards,

Bashar Abu Khalaf, PhD

Academic Editor

PLOS ONE

Journal Requirements:

2. In the online submission form, you indicated that your data is available only on request from a third party. Please note that your Data Availability Statement is currently missing the name of the third party contact or institution / contact details for the third party, such as an email address or a link to where data requests can be made. Please update your statement with the missing information.

4. We notice that your supplementary tables are included in the manuscript file. Please remove them and upload them with the file type 'Supporting Information'. Please ensure that each Supporting Information file has a legend listed in the manuscript after the references list.

Reviewers' comments:

Reviewer's Responses to Questions

**Comments to the Author**

1. Is the manuscript technically sound, and do the data support the conclusions?

Reviewer #1: Yes

Reviewer #2: Yes

2. Has the statistical analysis been performed appropriately and rigorously?

Reviewer #1: Yes

Reviewer #2: Yes

3. Have the authors made all data underlying the findings in their manuscript fully available?

Reviewer #1: Yes

Reviewer #2: Yes

4. Is the manuscript presented in an intelligible fashion and written in standard English?

Reviewer #1: Yes

Reviewer #2: Yes

Reviewer #1: 1). The placement of the organization of the study between the introduction and the theoretical framework disrupts the flow of information. I suggest to remove organization of the study to ensure a smoother transition and clearer progression of ideas.

2).To improve the logical flow of the introduction, start by providing an overview of the research topic, followed by defining key concepts. Then, link the study to relevant theories, highlighting the theoretical gap in existing literature. Next, identify the empirical gap, explaining the limitations of previous research in addressing your specific focus. Finally, clearly state the objective of the study, outlining its contribution to filling these gaps. This structure ensures a smooth progression from context to the research's purpose and significance.

3). After presenting the theoretical framework, it is crucial to include an empirical review that synthesizes key studies examining the effects of trade policies, particularly export controls, on corporate resilience and internationalization.

4.)It is recommended to include an extensive description of the data used in the study, detailing the sources, scope, and characteristics of the sample, while also discussing the potential benefits, such as the comprehensive coverage of A-share manufacturing companies, and limitations, such as potential biases or data gaps, to provide a clearer understanding of the data’s relevance and reliability.

4.)The findings would benefit from a discussion of potential causality concerns and omitted variable bias.

5.)It is recommended to provide a clear justification for the selection of control variables by explaining how each variable is relevant to the research question and how it helps isolate the effect of US export controls, rather than merely listing them, while also discussing why other potential factors were excluded, based on their irrelevance to the study’s focus or potential for introducing confounding effects.

6.) It is recommended to clearly state the source of each table within the paper, specifying whether the data is derived from primary research, publicly available datasets.

7.It is recommended to explicitly state the study's limitations, particularly regarding any data constraints, such as sample size, data availability, or potential biases, and address possible measurement errors that could affect the validity of the results, ensuring a transparent and balanced interpretation of the findings.

8.)It is recommended to include suggestions for further development of the field by exploring additional dimensions, such as the long-term impact of export controls on firm innovation, comparing the resilience of firms across different countries, and incorporating qualitative case studies to better understand strategic responses to trade disruptions, all of which would provide richer insights and help refine future research in the area.

Reviewer #2: The study "The Influence of the U.S. Export Controls Against China on the Resilience of Chinese Corporates" is topical and engaging, presenting the context of escalating U.S.-China trade frictions and how the new U.S. export control measures are affecting Chinese enterprises. Through the methodology employed, including the use of a multi-period difference-in-differences model and the analysis of data from listed companies in China's manufacturing sector, the paper highlighted the negative impact of controls on corporate resilience, identifying conditions under which companies can mitigate these effects.

My specific observations and recommendations are:

1. In the Introduction section: a) while mentioning the impact of export controls on corporate resilience, it should be clearly emphasized how this research differs from previous studies. What are the new or unique aspects of this approach in analyzing the impact of export controls; b) simplify some sentences that are too long to improve readability and clarity of the text; c) introduce some details on how and why these export controls have evolved over time, linking them to specific policy and technological changes, to provide a deeper understanding of the context.

2. In the Analysis of theoretical mechanisms and research hypotheses section : a) in section 2.1, explain why US export controls affect production efficiency and technological costs for Chinese companies, providing specific examples or case studies; b) diversify the source of studies cited to support the claims and increase the robustness of the arguments; c) introduce a discussion of long-term strategies that companies can adopt to address the negative effects of export controls, not just short-term responses or temporary adaptations.

3. In Section 3: a) why sales revenue growth and the standard deviation of monthly stock returns are most relevant for measuring corporate resilience; b) explain how each control variable was chosen; c) in describing the data selection process, what is the reason for excluding ST, *ST and PT firms to clarify how this affects the integrity of the sample.

4. In Section 4: a) Although the explanatory variables are said to be significantly positive, in Table 2, the DID coefficients are negative. This seems to contradict the statement that the impact is positive. This discrepancy needs to be clarified to properly explain the meaning of the impact of the variables on firms' resilience; b) When checking for robustness using the PSM-DID test, how were the variables selected for the propensity score calculation; c) The interpretations and conclusions drawn from the placebo test are not clearly related to the study objectives.

5. The "Further analysis" section exhaustively covers the mechanisms through which export controls influence firms' resilience. Suggestions and corrections to be made to improve the clarity and accuracy of the text: a) In the introduction of section 5.1, it is mentioned that export control "will reduce" instead of "reduces" or "has reduced", which would be more appropriate to reflect observed effects and not predictions or generalizations; b) The term "toughness" used to describe firms' resilience seems unclear. Perhaps "resilience" or "capability to withstand external shocks" would be more appropriate, as they are more specific and easier to understand; c) Table 5 describes the effects of the variables through the interaction with DID, but it would be useful to clearly explain how each of these variables is conceptualized and measured before presenting the results. For example, what do "Overfirm" and "Oversold" represent before discussing their results; d) When discussing political connectedness and strategic aggressiveness, provide more details on how these are assessed and why they are relevant to the study of resilience to export controls; e) In the analysis based on analyst attention and political connectedness, explain why these characteristics may influence how firms respond to export controls by also referring to recommendations from other countries and economies(https://doi.org/10.1080/21665095.2022.2032237;http://www.transformations.knf.vu.lt/53a/article/expo; https://doi.org/10.1515/ev-2023-0014; DOI: 10.15838/esc.2022 .6.84.4). How can analysts improve risk management or how political connections can facilitate access to resources in times of trade restrictions?

**Do you want your identity to be public for this peer review?** For information about this choice, including consent withdrawal, please see our Privacy Policy

Reviewer #1: No

Reviewer #2: No

---

## [Author Response · Author response to Decision Letter 1]

12 Jun 2025

Response to viewers

Dear Editor and Reviewers,

Thank you very much for your comments regarding our manuscript entitled "The Influence of the U.S. Export Controls Against China on the Resilience of Chinese Corporates". We are truly grateful to your critical comments and thoughtful suggestions on how to improve our manuscript. The valuable comments and criticisms have provided important guidance for our research and the revision of our manuscript. Based on these comments and suggestions, we have made appropriate modifications on the original manuscript, then we hope to meet your approval. The revisions themselves are addressed point-by-point in the subsequent text.

Response to Reviewer #1

1). The placement of the organization of the study between the introduction and the theoretical framework disrupts the flow of information. I suggest to remove organization of the study to ensure a smoother transition and clearer progression of ideas.

Response: We sincerely thank the reviewers for their valuable comments on the flow of information in the research organisation section of the paper. We fully agree with the suggestion to remove the research organisation section and have removed it at the end of the introduction in the first part of the revised paper. As shown in the manuscript (P6).

2). To improve the logical flow of the introduction, start by providing an overview of the research topic, followed by defining key concepts. Then, link the study to relevant theories, highlighting the theoretical gap in existing literature. Next, identify the empirical gap, explaining the limitations of previous research in addressing your specific focus. Finally, clearly state the objective of the study, outlining its contribution to filling these gaps. This structure ensures a smooth progression from context to the research's purpose and significance.

Response: Thank you very much for your professional comments on our article. According to your suggestions, we have optimized the structure of the introduction section and specifically modified it as follows (P2-P6) :

①To address the conceptualization issue, after outlining the topic and background of the study, we added conceptual definitions of export control and business resilience. In particular, the definition of export control adopts the relevant conceptual definitions in the U.S. Export Administration Regulations (EAR). Described as "Export controls, as delineated by the Export Administration Regulations (EAR)……" (P3).

The definition of firm resilience is based on Sajko's (2020) statement. Described as "Existing research has defined business resilience as an important ability of a business entity to cope with external shocks and secure long-term growth (Sajko, 2020). " (P5) .

②Regarding the shortcomings of existing literature, after elaborating the theoretical gaps in existing literature, we added the current limitations of empirical research. Described as "From the viewpoint of the current state of empirical research, although there has been a gradual increase in research on trade policy and enterprise resilience in recent years. …… " (P5) .

③With regard to the significance and contribution of the research, after explaining the limitations of previous studies on export control and firm flexibility, we sort out the purpose, contribution and significance of the research in the last paragraph of the introduction. Paper's contribution to filling existing research gaps is outlined in the innovation points and contributions. Described as "Compared to prior studies, this paper's potential innovations and contributions mostly lie in the following areas. ……" (P6).

3). After presenting the theoretical framework, it is crucial to include an empirical review that synthesizes key studies examining the effects of trade policies, particularly export controls, on corporate resilience and internationalization.

Response: We sincerely thank the reviewers for their valuable suggestions. We have expanded the Literature Review section by adding a literature review of relevant empirical studies (P5), which summarizes the key studies examining the impact of trade policy (especially export control) on firm resilience and internationalization. In addition, in the Theoretical Mechanisms Analysis and Research Hypotheses section, we add relevant studies on trade policy uncertainty, export control, and firm resilience (P7-P8) and add more references (references [23]-[35]) to support the article's ideas. Specific modifications are as follows:

①The literature review of the complementary empirical studies is presented as " From the viewpoint of the current state of empirical research, although there has been a gradual increase in research on trade policy and enterprise resilience in recent years. ……" (P5).

②The supplementary related literature research is as follows " Despite the gradual increase in research on trade policy and enterprise resilience in recent years ……" (P7-P8):

③Additional references are given. Described in the part of "References " (references [23]-[35]):

4). It is recommended to include an extensive description of the data used in the study, detailing the sources, scope, and characteristics of the sample, while also discussing the potential benefits, such as the comprehensive coverage of A-share manufacturing companies, and limitations, such as potential biases or data gaps, to provide a clearer understanding of the data’s relevance and reliability.

Response: We thank the reviewers for your careful review of the data description of this article. In response to your comment ‘need to add data sources, sample characterisation and discussion of strengths and weaknesses’, we have systematically improved the data section based on the original article. Described in the part of " 3.3 Data sources " (P18-P20).

① Address the issue of sample source. The US export control data in this paper comes from the official website of the US BIS and is manually sorted out, the original data of the enterprise's explanatory variables and control variables comes from the CSMAR database, and the mechanism variables and moderating variables come from the CSMAR, WIDN and other authoritative databases and are further calculated by the authors, and the data source of each variable in this paper is labelled in the table below. CSMAR and WIND databases are recognised as the most authoritative and comprehensive data sources in the field of China's capital market research, and their data collection process strictly follows the public information such as annual reports of listed companies and stock exchange announcements, and after multiple cross-checks, their reliability has been widely recognised by academics at home and abroad.

② Data scope and characteristics. This paper mainly focuses on the analysis of China's manufacturing enterprises, according to the ‘Guidelines for Industry Classification of Listed Companies (Revised in 2012)’ issued by the China Securities Regulatory Commission, the manufacturing industry category includes 31 categories from C13 to C43, including, but not limited to, machinery manufacturing, electronic equipment manufacturing, chemical industry, pharmaceutical manufacturing, etc., which is able to reflect the overall situation of China's manufacturing industry in a more comprehensive manner. It can reflect the overall situation of China's manufacturing industry in a more comprehensive way. According to the classification of listed companies in the CSMAR database, the sample of enterprises in 2015-2023 covers 29 major categories of enterprises in the manufacturing industry, while C16 (tobacco products industry) and C43 (metal products, machinery and equipment repair industry) are naturally excluded due to the absence of listed companies. The enterprise sample in this paper also covers data from 29 broad categories of enterprises in the manufacturing industry, essentially achieving comprehensive coverage of manufacturing enterprises.

③ To address the issue of potential advantages of the data, in the industry dimension, the sample of this paper focuses on the manufacturing industry, which is an important pillar of the national economy and occupies a key position in the national economy. At the same time, this paper takes Shanghai and Shenzhen A-share listed manufacturing companies as the research sample, which can reflect the situation of representative enterprises in China's manufacturing industry in a more comprehensive way; in the time dimension, the sample of this paper spans 2015 - 2023, during which China's manufacturing industry has experienced rapid development and transformation and upgrading, and at the same time, faces external challenges such as U.S. export control. The longer time span helps us observe the changes in firms' resilience in different economic environments and policy contexts, and capture the dynamic adjustment process of firms before and after the implementation of the export control policy; in terms of data processing, this paper takes a series of rigorous steps, such as excluding firms with financial anomalies, firms with consecutive observations less than three years old, and the samples with missing main variables, and conducting Winsor shrinkage on the continuous variable processing. These processing measures help to control the effects of outliers and data noise and ensure the quality of the data and the stability of the model estimation.

④ To address the issue of potential limitations of the data, despite our rigorous screening and processing of the samples, there may still be some bias. For example, this paper mainly selects A-share listed companies as samples, and there may be sample bias because these companies are relatively larger and have better governance structures, which may not fully represent all Chinese manufacturing enterprises, especially a large number of small, medium and micro enterprises. During data processing, we excluded samples with missing main variables, and missing data may lead to selective bias in the sample, affecting the generality and accuracy of the research results. In addition, due to the timeliness and availability of the data, we may not be able to fully capture all the factors that may affect firm resilience, which may lead to some bias.

5.) The findings would benefit from a discussion of potential causality concerns and omitted variable bias.

Response: We sincerely thank the reviewers for their valuable comments.

In response to the potential causality issue you mentioned, we would like to explain how the design of this study avoids the potential causality issue and add relevant discussions to enhance the rigour of the argument. The core explanatory variable in this study is U.S. export control policy (with whether an industry is included in the Entity List as a proxy variable), while the explanatory variable is the resilience performance of Chinese firms. The update of the Entity List is led by the Bureau of Industry and Security (BIS) of the U.S. Department of Commerce. In terms of the logic of policy formulation, decisions on U.S. export control policies are mainly based on macro-strategic considerations such as U.S. national security and technological competition, rather than the business conditions or resilience levels of individual Chinese firms. Meanwhile, this paper adopts the implementation policy as a quasi-natural experiment. Policy implementation comes before, and changes in enterprise resilience come after. Chinese firms are unable to influence the U.S. government's regulatory decisions by adjusting their own resilience levels in a reverse direction, which mechanistically excludes the possibility of reverse causation. Described in the part of "3.2.2 Explanatory variables" (P15).

In order to further discuss the issue of potential causality rigorously in terms of empirical research, we have taken three approaches to conduct robustness tests in the article. The first method is the PSM-DID, as described in the revised version "4.3.1 PSM-DID test" (P23-P24). The second method is the placebo test, which takes both the individual firm placebo test and the time placebo test, see the revised draft "4.3.2 Placebo test" (P24-P26) for details. The third method is to lag the control variables by one period for regression analyses, as detailed in the revised draft "4.3.3 Other robustness tests" (P26-P27). The results of the studies all prove that the impact of US export control policy on Chinese firm resilience is exogenous. See the revised draft (P23-28) for details:

The issue of omitted variables has been discussed in ‘4.3.3’. ‘In order to avoid estimation bias caused by omitted variables, this paper includes firms' invention patents and government subsidy variables in the regression equation.’ And it is found that the findings of this paper are still robust after considering the omitted variables, as detailed in "4.3.3 Other robustness tests " (P26-P28) of the revised version.

6.) It is recommended to provide a clear justification for the selection of control variables by explaining how each variable is relevant to the research question and how it helps isolate the effect of US export controls, rather than merely listing them, while also discussing why other potential factors were excluded, based on their irrelevance to the study’s focus or potential for introducing confounding effects.

Response: Thank you to the reviewers for their valuable suggestions. We fully agree with you about the control variables in the article. We have added the relevant content in the third point of the first part of the description of the measures and data in Chapter 3, Research Design, in the revised version of the paper. Described in the part of "3.2.3 Control variables" (P15-P18).

7.) It is recommended to clearly state the source of each table within the paper, specifying whether the data is derived from primary research, publicly available datasets.

Response: We thank the reviewers for their valuable suggestions. We have labelled the data sources below all tables (Tables 1-9) in the revised version of the paper (where the data in Tables 2 and 3 are exactly the same as those in Table 1, and we will not repeat the description) (P19-P41).

8. �It is recommended to explicitly state the study's limitations, particularly regarding any data constraints, such as sample size, data availability, or potential biases, and address possible measurement errors that could affect the validity of the results, ensuring a transparent and balanced interpretation of the findings.

Response: Thank you to the reviewers for their valuable suggestions. We fully agree with you on the control variables in the article. We have added a fourth section on study limitations for clarification in Chapter 3, Study Design, in the revised version of the paper. Described in the part of "6. Conclusions and policy recommendations" (P42-43), as "We list a few limitations to this study……".

9.) It is recommended to include suggestions for further development of the field by exploring additional dimensions, such as the long-term impact of export controls on firm innovation, comparing the resilience of firms across different countries, and incorporating qualitative case studies to better understand strategic responses to trade disruptions, all of which would provide richer insights and help refine future research in the area.

Response: Thank you for the reviewer's valuable suggestions. In response to your suggestion of "deepening the research through more dimensions", we have carried out in-depth discussions and made additional improvements, and now we would like to explain the relevant modifications and reflections as follows:

① Provide recommendations for further development of the field by exploring additional dimensions. Following the reviewers' comments, this paper explores the impact of more dimensional factors on firm resilience. Considering the supply chain as the core link of firm's operations, the effectiveness of its structure and management directly determines the firms’ ability to adapt and recover in the face of external shocks, this paper adds a comparative study of the resilience performance of firms with different supply chain concentration in coping with export control.

---

## [Decision Letter · Decision Letter 1]

13 Aug 2025

The Influence of the U.S. Export Controls Against China on the Resilience of Chinese Corporates

PONE-D-25-02339R1

Dear Dr. Huang,

We’re pleased to inform you that your manuscript has been judged scientifically suitable for publication and will be formally accepted for publication once it meets all outstanding technical requirements.

Kind regards,

Bashar Abu Khalaf, PhD

Academic Editor

PLOS ONE

Additional Editor Comments (optional):

Reviewers' comments:

Reviewer's Responses to Questions

**Comments to the Author**

Reviewer #3: All comments have been addressed

Reviewer #4: All comments have been addressed

2. Is the manuscript technically sound, and do the data support the conclusions?

Reviewer #3: Yes

Reviewer #4: Yes

3. Has the statistical analysis been performed appropriately and rigorously?

Reviewer #3: N/A

Reviewer #4: Yes

4. Have the authors made all data underlying the findings in their manuscript fully available?

Reviewer #3: Yes

Reviewer #4: Yes

5. Is the manuscript presented in an intelligible fashion and written in standard English?

Reviewer #3: Yes

Reviewer #4: Yes

Reviewer #3: (No Response)

Reviewer #4: The paper has been carefully revised and recommended for publication based on the reviewers' comments.

**Do you want your identity to be public for this peer review?** For information about this choice, including consent withdrawal, please see our Privacy Policy

Reviewer #3: No

Reviewer #4: No

---

## [Editor Report · Acceptance letter]

PONE-D-25-02339R1

PLOS ONE

Dear Dr. Huang,

I'm pleased to inform you that your manuscript has been deemed suitable for publication in PLOS ONE. Congratulations! Your manuscript is now being handed over to our production team.

Kind regards,

on behalf of

Dr. Bashar Abu Khalaf

Academic Editor

PLOS ONE